# Counting with Cilia: The Role of Morphological Computation in Basal Cognition Research

**DOI:** 10.3390/e24111581

**Published:** 2022-10-31

**Authors:** Wiktor Rorot

**Affiliations:** Human Interactivity and Language Lab, Faculty of Psychology, University of Warsaw, 00-927 Warszawa, Poland; w.rorot@uw.edu.pl

**Keywords:** morphological computation, basal cognition, morphogenesis, minimal cognition, cybernetics, mechanistic philosophy

## Abstract

“Morphological computation” is an increasingly important concept in robotics, artificial intelligence, and philosophy of the mind. It is used to understand how the body contributes to cognition and control of behavior. Its understanding in terms of “offloading” computation from the brain to the body has been criticized as misleading, and it has been suggested that the use of the concept conflates three classes of distinct processes. In fact, these criticisms implicitly hang on accepting a semantic definition of what constitutes computation. Here, I argue that an alternative, mechanistic view on computation offers a significantly different understanding of what morphological computation is. These theoretical considerations are then used to analyze the existing research program in developmental biology, which understands morphogenesis, the process of development of shape in biological systems, as a computational process. This important line of research shows that cognition and intelligence can be found across all scales of life, as the proponents of the basal cognition research program propose. Hence, clarifying the connection between morphological computation and morphogenesis allows for strengthening the role of the former concept in this emerging research field.

## 1. Introduction

The term “morphological computation” can be most generally defined as “computation obtained through interactions of the physical form” (the following exposition of the history of the term follows the one proposed by Paul [1]). Developing from the study of embodiment as a key property of intelligence (e.g., [2]), it has first been understood in a broader way as the interaction of physical form and the complexity of the control of a system [3]. Eventually, itv emerged in the full-fledged form of “morphological computation” in the work of Paul [1] and quickly took off among engineers and artificial intelligence researchers (see, e.g., [4] and the special issue of *Artificial Life* 19(1)) interested in “unconventional” computing platforms and intersecting with the study of soft robotics. The importance of this concept stems from the fact that it offers an important connection between the popular computationalist view on the nature of the mind and intelligence and the emerging and quickly gaining popularity (at the beginning of 21st century, currently well-established) anti-computationalist perspective focusing on the embodiment of cognitive systems (e.g., [5]). Morphological computation indicates that, in fact, accepting the importance of physical bodies does not preclude the possibility of conceiving the systems under study in computational terms.

However, together with the increasing popularity of the term “morphological computation”, there has been a growing number of instances where it has been applied only loosely. Those conceptual ambiguities prompted a response from Müller and Hoffmann [6], who have scrutinized the existing uses of the term and offered a critical perspective, significantly limiting the scope and applicability of the concept of morphological computation. Despite the importance and the great value of Müller and Hoffmann’s paper, the authors accept a restrictive perspective, which significantly reduces the role of this concept in the debate on embodied cognition.

In the current paper, I wish to suggest that, similar to how counting on fingers has become a paradigmatic example of embodied cognition in complex (‘higher’) mental phenomena, (see [7]), “counting with cilia”—morphological computation processes performed by single cells, such as those involved in morphogenesis—can be considered a crucial example in the debate considering the possibility of so-called computational enactivism [8] (see also [9]). However, this is only possible if we have a more permissive view on morphological computation than Müller and Hoffmann—a view that naturally arises from the mechanistic approach to computation.

As a sidenote, it must be clarified that “counting with cilia” is meant here primarily as a metaphor, as cilia’s involvement in morphogenesis is not fully known, and this phrase does not aim to suggest otherwise. For example, morphogenetic movements use primarily contraction and cell-to-cell adhesion [10], although an increasing body of research investigates the role of cilia for organ development due to their role in human disease (see, e.g., [11]).

The goal of the paper is to provide three contributions to the debate on the concept of morphological computation begun by the influential paper of Müller and Hoffmann [6]. First, I want to highlight that the definition of computation these authors use in their analysis is, in fact, a semantic definition, even though they do not acknowledge this fact. Indeed, if their arguments are viewed from the widely accepted perspective of the mechanistic view of computation, the distinctions and criteria they introduce become untenable and refer to properties that are irrelevant for “computation”. Second, in light of this analysis, I want to provide an alternative set of criteria, building also on the insights of the cybernetic movement. Finally, the paper introduces a novel example of a system that can be considered to be performing morphological computation: the developmental and regenerative morphogenesis in living organisms. I analyze this process with the criteria proposed by Müller and Hoffmann [6] and the ones introduced here, and point out how my perspective offers important advantages for this research. These goals are realized in Section 3, Section 4 and Section 5, respectively. Section 2 introduces the argument made by Müller and Hoffmann [6]. In Section 6, I make a case for linking the analysis of morphogenesis with morphological computation to the debates in the basal cognition research community and suggest what role morphological computation can play for computational enactivism. In Section 7, I conclude by reiterating the key points of the paper.

## 2. Semantic View of Morphological
Computation

Müller and Hoffmann [6] begin their argument with an overview of the approaches to physical implementation of computation. While this is an important part of the debate on the nature of computation, the question of implementation of computations in physical systems is partially orthogonal to the distinctions relating to the nature of computation in general. More precisely, the semantic view, at least in some versions (e.g., [12]) allows for both semantic and non-semantic perspectives on implementation, although the alternative mechanistic view, with its prohibition on involving semantics in computational descriptions, requires a non-semantic approach to implementation. Importantly, the non-semantic approach to implementation refers only to the criteria of applicability of a computational description to a physical system; it does not preclude the possibility that the physical system in question processes semantic information.

The semantic view of computation has a long history, building on early work by Hilary Putnam (e.g., [13]) and has been initially advanced by Jerry Fodor [14]. Although there are different varieties of the semantic account, they all accept Putnam’s premise of “no computation without representation” [15]. The semantic view requires that for a physical system to be counted as a computer, it must meet two criteria: (1) there must be a mapping defined from the physical representational states of the system to states defined by a computational description (e.g., states of a Turing machine), and (2) the state transitions between physical states must be causal and equivalent to transitions between computational states [15]. The details of these criteria, e.g., the individuation of representations and the sources of their semantic content, differ between varieties of this approach. For example, in a very recent iteration of the semantic view, Shagrir (pp. 179–181 of [12]) proposes to define the content in terms of “aboutness” and takes a pluralistic stance towards its origins.

Müller and Hoffmann [6] begin their analysis by accepting a particular view of computational implementation put forward by Horsman et al. [16]. This definition sets criteria that a physical process must meet to be counted as implementing “real, full-blown” computation:

An abstract initial state $m_{p}$ is encoded into the physical system using $I_{T}\left(m_{p}\right)$, giving rise to p; the physical system is left to evolve, reaching $p^{\prime }$, which is finally decoded using $R_{T}\left(p^{\prime }\right)$, giving rise to $m_{p^{\prime }}$, which we are assuming corresponds to $m_{p}^{\prime }$—the desired outcome of the abstract evolution. In this case, the physical system has acted as a computer [6], p. 9.

*T* refers to a theory of the physical computational device, which encapsulates the correspondences between the abstract computation and the physical computer and a description of the evolution of the physical states. Further, Müller and Hoffmann follow Horsman et al. in specifying that the representation relation they posit is one that a user introduces: “the occurrence of representation is a vital part of physical computing, and computational entities (human or otherwise) are the ones performing it” [16], p. 15. I wish to argue this clearly places their definition of physical computation within the scope of the semantic view of computation, although this is not explicitly acknowledged. This is because this definition claims that physical systems perform computation only if there is an objectively identifiable representation relation between physical and abstract states that is “performed” or introduced by a particular privileged “computational entity”: the user. Horsman et al. clarify with an example of hieroglyphic inscriptions: in their view, between the loss of the language and the deciphering of the Rosetta Stone, such inscriptions were just *potential* communications, and only once they were decoded did they became communications in actuality. Similarly, a physical system is only *potentially* a computer until a particular user imposes a representation relation on it, individuating the computations performed and making it an actual computer. The representation relation in their view has, in fact, a semantic nature: it is an abstract description of a physical object [16], p. 4. As such, it goes beyond the accepted non-semantic views of representation, which usually begin simply with syntactic information. Hence, in this view, computational individuation hinges upon the existence of a semantic relation between physical and abstract entities.

Müller and Hoffmann further require that the theory *T* be known a priori. This last criterion has been introduced originally by Copeland [17] as one of the criteria for suitable mappings between abstract computations and physical computers: Copeland states that such mappings should not be constructed ex post facto. However, as Miłkowski [18], p. 33 convincingly argues, referring to cases of reverse engineering of clearly computational systems (in which such mapping is only available posterior to empirical research), this restriction is not well-defined.

This semantic view underlies Müller and Hoffmann’s analysis of morphological computation. Scrutinizing cases discussed in the literature in those terms, most importantly: the (active) passive–dynamic walker (e.g., [19]), the eye of a fly (e.g., [20]), and examples of physical reservoir computing, such as mass–spring systems (e.g., [21]), they come to the conclusion that these systems fall into three classes:

**A. Morphology facilitating control**: In systems such as the passive–dynamic walker, they argue morphology is better seen as simplifying the controller or enabling particular actions that otherwise would be impossible. Their analysis indicates that, in these cases, there is no representational relation between abstract computation and the morphology of the system—no suitable encoding and decoding can be defined—and the theory of the system does not indicate that its evolution tracks the evolution of the abstract computation.

**B. Morphology facilitating perception**: In systems such as the eye of the fly, morphology can be associated with a proper computational description: we know well what transformation the morphology of the system performs and can describe it as an input–output (computational) mapping—hence, the evolution of physical states tracks that of computation. However, or so Müller and Hoffmann would argue, the morphology is not “deployed here as a computational substrate to calculate an abstract problem” [6], p. 17, an observation that probably comes from the difficulties with identifying a proper “user” of such a system, who could define the representational mapping between abstract and physical states.

**C. Morphological computation**: Eventually, systems such as physical reservoir computing constitute the class of systems that can accurately be described as morphological computers, according to Müller and Hoffmann [6]. Such systems are employed in human-designed computing frameworks with controlled inputs and outputs and known dynamics of the system. As a result, such systems clearly meet the criteria introduced above.

## 3. The Mechanistic Alternative to the Semantic
View

The difficulties of defining and individuating representations and of naturalizing their content ultimately led many philosophers, especially within the philosophy of cognitive science, to increasingly distance themselves from the semantic view. In the result of the emergence of the new mechanism philosophy [22], the main competing view has crystallized in the form of the mechanistic account of computation (cf. [18,23,24]). The mechanistic view defines physical computation as processing of vehicles according to rules sensitive to some properties of the vehicles. Computational vehicles are symbols over which the computation is performed. These are encoded in a particular physical medium, but most theories of computation take this medium to be relevant only insofar as computational rules are specific to differences between different portions of the vehicle along one dimension of variation (single physical property). For example, in the case of neural computation, neuronal spike trains are the vehicles, but computation is often taken to refer to their medium-independent properties, such as spike rates. Crucially, processing is performed by a mechanism that is functionally organized to perform this computation. Here, different varieties of the mechanistic account will differ, among others, by the view of function they accept. The concept of function, although no less problematic than that of representation, is, however, more basic; hence, difficulties with establishing its naturalized definition seem to deter philosophers to a lesser degree.

Proponents of the mechanistic view reject the requirement of encoding/decoding or representational relation between an abstract object and a physical one. Instead, computation under this view corresponds to any instance of information processing, where one can distinguish a particular mechanism that performs information processing as its function. The identification of computation with information processing (although at times contended within the mechanistic view, see Piccinini [23]) refers to purely syntactic information without presupposing its representational character. In particular, Miłkowski [18] draws on MacKay [25] and makes use of the concept of “structural–information–content”, which is ontologically “cheap”, as any physical process that has at least one degree of freedom (independently variable property) will carry some structural information (given this very precise sense of information, Piccinini [26], in fact, concedes that computation is always information processing).

Miłkowski [18], pp. 77–78 explicitly touches upon the topic of morphological computation. He argues that given a mechanism that is (evolutionarily) designed to perform the function of processing information and that can be clearly individuated (as is the case in biological systems, with delineated cells, tissues, etc.), the computational function can be ascribed to it. It does not matter whether this function is performed thanks to the morphological features in a physical, analogue way, or thanks to some other (possibly medium-independent) properties of the system. Miłkowski builds on the example of the mechanism of the pressure-difference receiver in crickets cf. [27], a case resembling that of the eye of the fly—so a case of morphological perception in Müller and Hoffmann’s framework. This view can be also expanded to provide a criterion to differentiate morphological computation from other types of computation: in the morphological case, computational vehicles are processed in a medium-dependent way, with computational rules being sensible to their physical properties. In the case of the pressure-difference receiver, the criteria are met, and a computational description in terms of sound filtering can be successfully ascribed to the system. This is, however, more than a minimal restriction that (some flavors of) the mechanistic account requires: in fact, all cases of sensing can be counted as computation, as they all involve information processing (the minimum would be for them to compute an identity function, although in physical systems, it is more complex as the information is most often transduced and digitized, either into bits or into nerve cell firing patterns—so it undergoes a nonlinear transformation).

Just as in the eye of the fly, the pressure-difference receiver in crickets falls neatly under the scope of “intrinsic computation” in Müller and Hoffmann’s framework. In their view, intrinsic (natural) computation is classified as “real” computation only if there is a user, the encoding and decoding processes are designed, and the computational description can be applied a priori (i.e., only the criterion of having a suitable theory describing evolution of states is met). However, under the mechanistic view, these additional criteria are not relevant, and nothing excludes the examples of “intrinsic” computation out of the realm of “real” computation.

Hence, the tripartite classification of Müller and Hoffmann [6] introduced above under the mechanistic view must be simplified. Above, we have seen that simply accepting the mechanistic criteria of computation dissolves the distinction between morphological perception and morphological computation (for clarity, I will refer to this class of processes simply in terms of morphological information processing as, for practical reasons, it may at times be relevant to distinguish between these different classes of computational processes). I turn now to the discussion of the other class of processes: morphological control. In that endeavor, I will refer to the cybernetic tradition to introduce key distinctions.

## 4. Cybernetics, Computation, and Morphological
Control

The cybernetic tradition, in particular the so-called first wave of cybernetics, has influenced both computationalist and anti-computationalist views in the scientific study of mind, philosophy of mind, and—by extension—in philosophy of biology (see for overview [28,29]). Froese [30,31] points out that these two veins of evolution resulted from one crisis in the original cybernetic thought, related to the emergence of W. Ross Ashby’s work, in particular, his Homeostat [32] (although the treatments in [28,29] do not assign as great of an importance to this point). The Homeostat presented a proof-of-concept for the design of an ultra-stable system, which through simple controllers is able to maintain a stable pattern of activity even in the face of major perturbations of the environment (e.g., a reversal of polarity of the phenomenon it is supposed to control) through a change of internal organization or structure. Ashby did not see the tension between the design of the Homeostat, a dynamical system through and through, and the notion of computation, as he himself calls it an “analogue computer” [32], p. 99, which constantly searches the parameter space to maintain its computation of a (more-or-less) constant function in the face of changing inputs. Interestingly enough, in Froese’s analysis, the appreciation of the insights coming from the Homeostat eventually led to the emergence of enactive cognitive science and of second-order cybernetics. The former view adamantly rejects computationalist explanations of cognition, disregarding Homeostat’s computational nature underscored by Ashby.

Importantly, there is largely an agreement among commentators (e.g., [28]) that the original perspective on computation of cyberneticians falls under the umbrella of the mechanistic view. However, the important contribution of cybernetics that I want to build upon here concerns specification of the concept of “control” in terms of feedback loops.

The argument for keeping “morphology facilitating control” as distinct in the mechanistic framework of morphological computation could be made by reference to the seminal concept of “intentional stance” introduced by Dennett [33]: it is simply not necessary to invoke information processing to describe the operation of, e.g., a passive–dynamic walker. The concept of “intentional stance” was proposed by Dennett to account for ascription of mental states to various systems, including ourselves and other human beings. Dennett proposed that whereas some systems can appropriately be described from a physical or “design” perspective or stance, referring to their dynamics or designed functions, respectively, some require a more abstract account. This is the case with humans or other animals, where we decide to ascribe mental states—beliefs, desires, intentions—to be able to explain and predict their behavior. What I mean here, in the context of “morphology facilitating control”, is that whereas we might decide to describe the movement of a passive–dynamic walker by referring to the concept of information (e.g., information about the steepness of the incline being processed to establish appropriate speed of movement), a less abstract description, in fact, works better (e.g., in terms of forces and classical mechanics). However, the argument along these lines that Hoffmann and Müller [34] originally offers is not entirely correct, as they invoke the sufficiency of dynamical systems’ description for these kinds of systems but ignore the fact that those can be also used for computational systems [35]. In fact, they are the primary way to describe the operation of analogue computers (formally speaking, they both involve differential equations to describe the processes; see, e.g., [36]). Nonetheless, they are correct in the fact that a physical description (i.e., Dennett’s physical stance) pitched at the level of classical mechanics suffices for systems such as the passive–dynamic walker, and no reference to informational notions is needed for this purpose.

However, even using the term “control” to refer to the passive–dynamic walker is not so obvious: its behavior is, in principle, no different from that of a billiard ball rolling down an incline, and one would not invoke the term to describe the ball (in fact, unless the incline somehow constrains the movement of the ball, one may be more likely to call it uncontrolled). While a feedback loop (see [37], p. 12) is not a necessary condition of control—we would say that a person who simply presses the gas pedal while keeping the steering wheel still controls the vehicle (provided the vehicle operates normally and no spurious relationships are in play), even if s/he does so terribly and dangerously—“feedback” does delineate a particularly interesting subset of control processes. Feedback loops were initially defined by Wiener (see also [38]) by reference to a number of canonical examples, such as a thermostat or a watt governor (for an important case study in philosophical debates on the role of representation for cognition, see, e.g., [39], controlling the operation of a steam engine). Essentially, a feedback loop provides an informational relation between the output and input of a process through which the process can be better adjusted. To return to the car example: as soon as the driver opens his/her eyes, s/he will be able to relate the input (pressing of the gas pedal) and the output (a quickly approaching concrete wall) and act on the input to return the system into viable conditions (stop pressing the gas pedal and bring the car to a stop). Importantly, Conant and Ashby [40] distinguish between “error-controlled” and “cause-controlled” regulation. These can be illustrated with an example from Ramsey [41]: consider two drivers following an S-shaped road in cars that lack windows so that the drivers cannot actually see where they are going. Car A is equipped with a proximity sensor that informs the driver when s/he is approaching the edge of the road. Car B is provided with an accurate map, which the driver uses along with a simple dead-reckoning strategy to guide his/her way along the road. This is contrasted with a third car, Car C, that lacks any sort of driver and bounces off the railing. All three cars manage to move along the road despite their differing strategies. The first two cases (Cars A and B) correspond to error-controlled and cause-controlled regulation, respectively. What is relevant for current purposes is that in both cases a sort of information processing is required. In the case of Car C, no informational processes are involved; the movement of the car proceeds similar to the movement of “a marble in a pinball machine” [41], p. 194—not so different from a passive–dynamic walker.

This brief discussion shows that the class of cases that could be classified as “morphological control” is heterogeneous, and the examples provided by Müller and Hoffmann [6] fall mostly under the category that one may call “not even control”. Consider again the primary example of a passive–dynamic walker: its walk down an incline is no different (in terms of a control) than a billiard ball rolling down the incline or a marble in a pinball machine bouncing off the edges. If components of control are present in those systems (e.g., in the passive–dynamic walker with actuators), they do not perform the control function in virtue of their morphology but rather in a more classical way.

However, if we look at examples from the class “control with feedback” (comprising both error-controlled and cause-controlled examples), it turns out that there are systems that perform control in virtue of their morphology and qualify as computational according to the mechanistic criteria. The examples of the thermostat and the watt governor can be brought up here: these simple systems compute very basic functions (“less than”), and they do so in virtue of their structural and morphological properties (the shape or size of a metal bar in a classical thermostat, the angle between the arms in the watt governor). Although obviously these systems can also be defined with simpler, physical descriptions, these computational ascriptions seem sufficiently warranted.

As it seems reasonable to assume that there are some systems that would qualify as “morphological control without informational input” (although no examples can be found in Müller and Hoffmann [6], and I was unable to provide any as well), then we can take the sets of systems exhibiting morphological control and performing morphological computation as intersecting. None of the examples that Müller and Hoffmann provide, however, qualify as even morphological control under the criteria proposed here: while the passive–dynamic walker has been discussed extensively above, we may also recall the example of gecko feet. Müller and Hoffmann argue that gecko feet, which have a particular hierarchical morphology enabling the appearance of appropriate van der Waals forces between the feet and vertical surfaces, allowing geckos to climb even smooth surfaces, are a case of morphological control. However, the adhesion of feet to surfaces is not control, just as the fact that a car hook’s shape, which enables it to attach a looped line and tow other vehicles (to remain in the realm of automotive examples), is not a form of control on its own. Control in the case of gecko feet relates to the forces the animal applies to adhere the feet to the surface and to move—and this is not performed morphologically.

Hence, under the cybernetic–mechanistic view accepted here, the tripartite distinction collapses into a two-fold one, although in some cases the original classes provide a good taxonomy of forms and purposes of morphological computation (see Figure 1 for a diagrammatic depiction).

## 5. Morphological Computation in Developmental and Regenerative
Morphogenesis

The study of morphogenesis can be dated back to Aristotle [42], but in its modern version, it began with the mathematical and physical descriptions proposed by [43] and Turing [44]. Morphogenesis is the process responsible for the development of the shape of biological structures (cells, tissues, and organisms). While all biological systems undergo this process during development, many are capable of it throughout their lifetime, and hence, we can distinguish between developmental and regenerative morphogenesis, although their mechanisms are speculated to be largely identical.

Importantly, quite early attempts have already cast morphogenesis in terms of computation. One of the earliest explicit statements of this perspective comes from Wolpert and Lewis [45], p. 21, which attributes the following question to Sydney Brenner: “Is the adult (organism) computable from an egg?” This line of research has been expanded by, among others, Paulien Hogeweg (e.g., [46]), and more recently by Michael Levin and his collaborators (e.g., [47]), whose work is analyzed in more detail below. However, the computational nature of this process was already implicit in the reaction–diffusion system proposed by Turing, as it has been shown since that such systems can constitute analogue computers [48]. Hence, the mechanisms implementing morphogenesis in biological systems are at least partially computational in nature (according to the definition of computation accepted here, or “intrinsically computational” in terms of Müller and Hoffmann [6]).

There is, however, a discontinuity in this research tradition. Brenner, Wolpert, and Lewis, when they asked whether an organism is computable from an egg [45], p. 21, were interested if a researcher, given the genetic information in the egg, would be able to “compute”, i.e., predict, an adult organism phenotype from it. They proposed a model of the morphogenetic process in terms of cellular automata governed by a genetic program implemented by binary gene switching networks. However, their account is metaphysically modest and makes no claims for the “computational nature” of the process thus described. Similarly in the work of Hogeweg: she has proposed computational models and simulations of morphogenetic processes (e.g., [46,49]) thath ave been described as attempts at computing an organism [50]. However, whenever she references the word “computing” or calls the morphogenetic processes “informatic”, she uses “scare quotes” around these words, differentiating computing from ‘computing’. These implicit assumptions can be interpreted in instrumentalist terms: Hogeweg uses the language of ‘computation’, computer simulations in particular, as it offers a good explanatory framework. She does not presuppose, however, that these explanations translate into actual computations in nature but rather into mechanisms that can be partially described as “processing information”.

Levin and collaborators’ work (e.g., [47,51,52,53]) introduces a major shift in this regard. Although he states that his work on morphogenesis follows the tradition discussed in the previous paragraph, he does not merely use the computational language for explanation but rather claims that the processes of morphogenesis are literally computational and process information. This is clearly visible in the way they update the aforementioned quote originally due Brenner: instead of asking how can we, scientists, compute an organism from an egg, Manicka and Levin ask “How does an embryo compute *its own* pattern?” [47], (p. 1; emphasis added). They further strengthen this perspective by exploring the possibility of using insights from the study of morphogenesis in the development of artificial computational platforms. Contrasted with Hogeweg’s instrumentalist view, Levin’s metaphysical perspective on scientific concepts can be best accounted for by the model-based philosophy of science (e.g., [54,55,56]), and his claims about computations can be understood as identifying robust properties [56] of the natural world.

Development and regeneration require information processing to meet several intertwined goals: patterning requires the cell to process (relative) locational information to move to an appropriate part of the growing organism; cytodifferentiation also requires input of positional information together with a model of the general body plan for developing appropriate cell types in appropriate places; finally, the dividing cells need to solve the most difficult problem of tissue growth: they require information about the structure of the organism attained thus far to be able to stop divisions when the correct shape is reached. While some of this information is encoded genetically (e.g., available cell types), existing experimental work shows that genetic information is hardly the main input into this computation process. Levin’s experimental work provides two important lines of argument in this regard.

First, while traditionally DNA is taken to provide information about shape, Levin’s work in planaria, which have astonishing regenerative capabilities and can regenerate a whole organism from even the tiniest slice, proves that much of the information is, in fact, encoded in bioelectric signals between cells (e.g., [57,58]; see also the review in [53]). Without altering the genome, Levin and collaborators induced bioelectrical changes by modifying ion channel expression in regenerating worms. This intervention modified the patterning so that the organism regenerated a second head instead of a tail (a body plan with 0, 1, or 4 heads can be thus induced, or even a head morphology of a different species, i.e., a flat head in a round-head worm species or vice versa). Furthermore, this new body plan became encoded in the bioelectrical activity, so that after further incisions, the new two-heads-no-tail body plan was retained without any further intervention [58].

Second, DNA cannot be set to encode a set of morphogenetic movements, or steps, but rather an “error minimization process” or a homeostatic loop [53], p. 15 (see also the work exploring connections of this approach to the free energy principle framework: [59]; [60]). The developmental and regenerative processes are highly robust, in that they are able to attain the correct result from a variety of abnormal starting conditions. Levin’s lab has shown this by modifying developing tadpoles. After scrambling the position of their facial organs, the developing frog alters its developmental ‘program’ to achieve the correct face shape [61]. Even more astonishingly, a tadpole created with an eye in an incorrect position (i.e., located on the tail) is still able to develop the eye correctly, together with the optic nerve that connects in a novel location to the spinal cord, allowing the tadpole to see quite well [62].

Both these examples show that morphogenesis is not a ‘blind’ feedforward process but rather an adaptive process that achieves “fixed ends with varying means”—what William James has proposed as a minimal criterion for mentality [53,63]. Hence, this research casts morphogenesis as a complex information processing and control process implemented by—and over—an ever-changing medium. This interaction of structure and information processing clearly puts morphogenesis in the domain of morphological computation, involving both morphological perception and morphological control. At the same time, this perspective introduces restrictions regarding the admissible concept of computation that it can be understood with: interpreting it through the lens of the semantic approach, particularly in the flavor proposed by Müller and Hoffmann [6], would require identifying semantic representation of abstract computational states in cell states and specifying a user of those computations. This is not the case: the computations are performed online by individual cells, and they are used by those very cells. However, the computations cannot be individuated on this level. Consider the idea of a “morphogenetic field” [64] on which Levin builds (e.g., [65]): this model takes morphogenesis to be guided by non-local interactions of cells (physiological or chemical gradients, steady-state bioelectrical properties, etc.), suggesting that morphogenesis is best understood not on the level of a cell, but rather at higher levels of organization. As a result, each cell does not need to represent its position, but rather it responds to available information, giving rise to the computational process at a higher level of organization. The “computational entity” or “user”, if we were to identify one in the case of morphogenesis, would need to be considered an emergent property of the process, appearing on that higher level of interacting cells. As such, it is impossible to establish a relevant encoding/decoding relation between physical and abstract states. The criteria of computation are not met under the semantic view.

However, under the mechanistic perspective, morphogenesis can be considered an example of computation: the cells process available information (in the form of available physiological or chemical gradients, bioelectrical properties, etc.) and use them to guide their actions. Further, as we have seen from the above discussion of the research on this biological phenomenon, the complexity of these processes exceeds what can be accounted for on the level of physicochemical interactions, requiring scientists to switch to the “informational stance” (cf. [33] and Section 4). Hence, under the mechanistic view, the more realistic approach of Levin and collaborators is, in fact, warranted.

## 6. Links to Basal Cognition

In the introduction, I briefly summarized the history of the concept of “morphological computation”, highlighting its roots in the embodied traditionally anti-computational view of the mind. The emerging research field of basal cognition is an intellectual descendant of this tradition. Basal cognition focuses on “the fundamental processes and mechanisms that enabled organisms to track some environmental states and act appropriately to ensure survival (finding food, avoiding danger) and reproduction long before nervous systems […] evolved” [66], p. 4. As such, it “describes a toolkit of biological capacities involved in becoming familiar with, valuing and exploring, exploiting or evading environmental features in the furtherance of existential goals” [66], p. 5. As a field, it aims to strengthen the inclusion of life science and biology—in particular, developmental biology and regenerative medicine—into the cognitive sciences. Basal cognition, similarly to the embodied framework in the late 20th century, aims to expand our understanding of cognition and of mechanisms that may be responsible for cognitive faculties by pointing not only to unconventional cognitive media but also to hitherto overlooked phenomena. Developmental and regenerative morphogenesis is one of the primary examples proponents of this field refer to. As already mentioned in the previous section, the adaptive nature of morphogenetic processes seems to meet classical criteria of mentality. In more contemporary terms, Levin [53], p. 2 expands James’ notion by proposing to understand intelligence as the degree of competence in navigating spaces. When thinking of the physical, 3D space, this definition includes processes we normally conceive of as intelligent (e.g., rats’ ability to navigate mazes) or that are more controversial (e.g., slime molds’ and oil droplets’ ability to navigate mazes; see [67,68]). However, this notion can also be expanded to other “spaces”: Levin mentions primarily “morphospace”—the space of possible morphological configurations of an organism explored in the process of morphogenesis, which makes morphogenesis conform to this definition of intelligence (other spaces he lists include transcriptional spaces explored by mRNA, physiological spaces explored, e.g., by ion channels, etc.). Importantly, Levin regards intelligence as a continuum in this regard so that we can discuss systems possessing less or more intelligence, and hence need not treat rats and slime moulds as equivalent (as rats can be regarded as *more* competent in navigating spaces).

Morphological computation can play the role of a central conceptual hub relating the research in basal cognition to the investigation of human brains in computational neuroscience. In a way, I want to suggest that it can play a key role in understanding the principles of “scaling up” cognition in its “endless forms most beautiful” and complex (for further suggestions about “scaling up”, see [69]).

While the brain does not necessarily involve “morphological” computation, the achievements of computational neuroscience (see, e.g., [70]) as well as significant developments of artificial intelligence, in particular in machine learning and neural network architectures (e.g., [71,72,73]) secured a central position in cognitive science for the computational theory of mind. Hence, showing that basal organisms not only employ equivalent mechanisms (e.g., as is the case with ion channels in bacteria; see [74]) but also are capable of performing complex computational functions supports the analogies between human and basal cognitive capacities that basal cognition research ultimately tries to draw.

Furthermore, basal cognition draws heavily on the investigations within synthetic biology and soft robotics (e.g., [75,76,77,78]), which is also one of the domains that supported the emergence of the ideas of morphological computation (e.g., studies of octopus arm control; see [79,80]). This concept, then, appears to permeate the research tradition, even if hitherto its central role was not appreciated.

Importantly, despite the continued tensions between proponents of computationalist and anti-computationalist theories of mind, recently, a middle way has been considered by some researchers. Researchers such as Villalobos and Dewhurst [9] and Korbak [8] have attempted to reconcile the radically anti-computationalist, anti-representationalist view of the enactive approach to the mind with computationalist perspectives. This attempt resulted in a view that Korbak calls “computational enactivism”. Morphological computation allows us to associate basal cognition with this idea. Computational enactivism argues that the autopoietic requirements set by enactivism are compatible with, or in a stronger form—co-dependent on, the computations performed by certain physical systems, in particular living organisms. If we consider morphological computation in the context of processes such as morphogenesis, where it is tasked with implementing a particular homeostatic loop and overseeing this autonomous, self-organizing — autopoietic — process, it provides significant support for the “computational enactivist” position. Consider further the example of the Ashby’s Homeostat, mentioned earlier, which has served as the source of inspiration for the theory of autopoiesis [31]. In virtue of this, we can regard the computational nature of morphogenetic processes as a proof-of-principle of a simple cognitive system conforming to the propositions introduced within computational enactivism and hence offering a way out of the existing binary.

An argument similar in spirit may be found in a paper by Beer and Williams [81]. The authors develop computer simulations of simple agents that are evolved to perform a categorization task. They analyze these simulations using the tools of information theory and dynamical systems theory—tools that are traditionally connected with the deeply opposed theoretical strands in philosophy of mind and cognitive science: computationalism and the embodied mind theory, respectively. They conclude that these two forms of mathematical descriptions are precisely that—different mathematical languages that can be used to elucidate different properties of the systems under study. These languages are not at odds, and although they have different purposes, their descriptions are often consistent and complementary. Crucially, this arises from a methodological approach similar to the one suggested here, namely, from focusing on a “minimal” system that exhibits sufficient complexity of behavior to be considered under the cognitive lens but at the same time is simple enough to be amenable to fine-grained analysis, including empirical studies in the lab.

While my argument takes the conclusions of Beer and Williams’s work for granted and builds on the commonalities between information theory and dynamical systems theory, the analysis provided here aimed to show that two properties taken to be at odds, autopoiesis and computation, can, in fact, be identified in the same system as interacting and co-supporting. In this way, I suggest regarding morphogenesis—the metaphorical “counting with cilia”—as an initial example of how computational descriptions may be of use for accounting for autopoietic and self-organizing processes.

## 7. Conclusions

This paper discusses the concept of “morphological computation”, pointing out that the existing critical analysis of Müller and Hoffmann [6] relies on a particular definition of computation that implicitly falls under the scope of the semantic view of computational implementation. Accepting instead the alternative, mechanistic perspective on physical computation offers a significantly different classification of processes that can be considered to be performing morphological computation. In fact, under the mechanistic view, including some insights provided by cybernetics, the tripartite distinction Müller and Hoffmann introduce between morphological control, perception, and computation proper collapses into a simpler view that distinguishes a limited number of very trivial cases of non-computational morphological control from morphological information processing that is “really” computational and can involve control and perception as well as human-designed computation.

I further substantiate these claims by analyzing an example of morphological computation that has been overlooked so far in this literature. I refer to studies of developmental and regenerative morphogenesis that have been increasingly regarded as computational in nature, and I point out that the computations in question are, in fact, performed morphologically. This example connects the analysis back to the debates within cognitive science, philosophy of mind, and philosophy of biology, as it is one of the central cases studied under the label of “basal cognition”. Its importance can hardly be overstated, as it puts basal cognition at the center of the stalled debate between computationalists and anti-computationalists, providing important support for a recently proposed “middle-way” approach—computational enactivism—aiming to reconcile insights from these opposite traditions.

Hence, this discussion of morphological computation offers the potential of significantly influencing our understanding of cognitive systems and intelligence, helping the project of basal cognition and allowing it to make novel connections to established fields, such as computational neuroscience in particular.

## Figures and Tables

**Figure 1 entropy-24-01581-f001:**
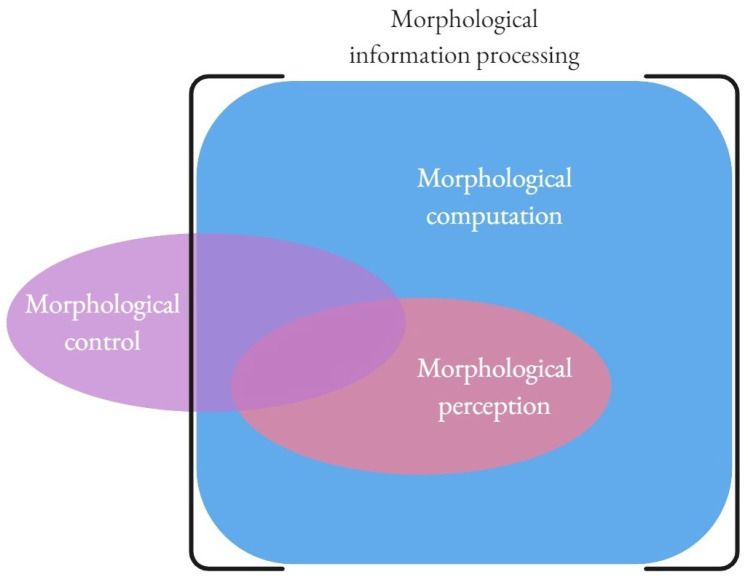
A diagrammatic illustration of the relations between the distinctions introduced by Müller and Hoffmann [6] under the cybernetic–mechanistic view of computation, as proposed here.

## Data Availability

Not applicable.

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
