# Peer review of "Counting with Cilia: The Role of Morphological Computation in Basal Cognition Research"

_entropy, 2022, doi:10.3390/e24111581_

Round 1

Reviewer 1 Report

The manuscript is substantially a reply to Müller and Hoffman (2017, A. Life) who suggested that the word “computation” in the term “morphological computation” was misleading. Those authors would have us believe that since the key examples of morphological computation (in animals and in robots) are not formal logical digital computation, they should not be treated as part of cognition, per se. However, it could be suggested that the flavor of computation that morphological computation is adding to cognitive processes is analog computation. By placing some emphasis on biological development and morphogenesis, the present article contributes novel important observations to this debate. With some minor revisions, I recommend acceptance.

Comments:

Somewhere in the manuscript, it could make sense to include some discussion of Randall Beer’s simulations of “minimally cognitive systems” (e.g., Beer & Williams, 2015, Cognitive Science).

When Section 3 introduces “vehicles”, it might be wise to provide a definition of vehicles for some readers.

In Section 4, when mentioning the Watt governor, it might be worth noting that Tim Van Gelder (1995, J. Philosophy) used the Watt governor as his “poster-child” for a representation-free dynamical control system.

In the last two paragraphs of section 4, it might be helpful to remind the reader of exactly how none of Müller and Hoffman’s examples would qualify as morphological control.

In Section 5, the second paragraph refers to an “explicit statement” attributed to Brenner about morphogenesis and computation, and a couple paragraphs later it is referenced as the “aforementioned quote originally due to Brenner.” However, the way these paragraph are written, I’m not sure I know what that explicit statement or quote from Brenner is. Perhaps it could be better foregrounded in the prose with quotation marks around it.

Minor comments:

In the third paragraph of Section 2, the author refers to himself as “we” but at the end of section 3, he refers to himself as “I”.  Please pick one and be consistent throughout the manuscript.

Halfway thru Section 3, there’s a sentence beginning with “Milkowski [13, pp.77-78] explicitly touches upon…” that is probably a run-on sentence. It might be good to find a way to carve that sentence into two sentences.

In general, the manuscript would benefit from a native English speaker combing through it carefully and correcting small grammatical errors.

Reviewer 2 Report

I  enjoyed reading this article verymuch and find it novel and refreshing contribution in current discussions on embodied cognition  and computational approaches.

There are a two minor remarks which I would suggest the author adresses.

1. The title is very attractive. But in  the text I could not find any mention of cilia or counting.  It  would be good to put it back as I believe  it was cutted our at some point.

2.  I would not say that Vincent Müller’s and Matej Hoffman’s distinctions are meaningless. They make sense for their purposes and under their premises, but if one is after unification, those distinctions do not matter. They are meaningful, but in this bigger context, they makes no difference. I react to the word “meaningless” which is repeated twice, and it feels harsh.

Author Response

Please the attachment.

Reviewer 3 Report

The article should be rejected because it contributes nothing to science. This is a review of some of the things done by other scientists, primarily Müller and Hoffmann, but also Milkowski. All cited materials are well known and all the conclusions drawn by the author are trivial. It is a student essay, not a scientific paper.

Author Response

While I appreciate the time the Reviewer has taken to read the paper, I must admit that this review is highly unhelpful.

As other reviewers have also pointed out, I believe that connecting the literature on morphological computation with research on morphogenesis does provide novel observations which move the debate on morphological computation forward. These connections may be of interest to an audience that has not hitherto taken part in these discussions. Even if we take in good faith the Reviewer’s point that cited materials are well known and the conclusions are trivial, I am unaware of any publication which would make the connections that the current paper is trying to make. Hence, it is difficult for me to see how my conclusions can be trivial. If the Reviewer is aware of any such paper, I would highly appreciate pointing it out to me. Otherwise, I believe that lack of publications which attempt to do the work of this paper plays the role of unnecessary gatekeeping, denying access to relevant knowledge to the part of the scientific community (in this case – researchers in biology, who are usually not included in the debates on morphological computation) for whom, differently than for the Reviewer, both this literature and the conclusions of the current paper are not obvious. In particular, I believe that the journal Entropy, with its interdisciplinary audience, is a unique venue for meeting the goal of including new groups of scientists in the discussions on morphological computation.

I also want to disagree with the point that the piece “contributes nothing to science.” Whereas this is a philosophical paper, which does not at any point aim to provide new data, I believe that philosophy of science, and in particular the so-called “synthetic philosophy” that serves as the methodological backbone of this work, serves a crucial role for scientific debates. This slightly external perspective can provide conceptual discussion of the key terms that scientists rely on in their empirical work, clarifying those terms and explicitly formulating their assumptions and connections. This has been attempted in the current paper, and, as other Reviewers have reported, I believe this task has succeeded at least to some degree.

At this point, I believe it is needless to say that I did not make any changes to the paper to incorporate the Reviewer’s comments.